


# The pathway of aerosol direct effects impact on secondary inorganic aerosol formation

Jiandong Wang[1,2], Jia Xing[1*], Shuxiao Wang[1], Rohit Mathur[3], Jiaping Wang[4] Yuqiang Zhang[5], Jonathan Pleim[3], Dian Ding[1], Xing Chang[1], Jingkun Jiang[1], Peng Zhao[6], Shovan Kumar Sahu[1], Yuzhi Jin[2], David C. Wong[3], Jiming Hao[1]

[1]State Key Joint Laboratory of Environmental Simulation and Pollution Control, School of Environment, Tsinghua University, Beijing, 100084, China
[2]Key Laboratory for Aerosol-Cloud-Precipitation of China Meteorological Administration/School of Atmospheric Physics, Nanjing University of Information Science and Technology, Nanjing, 210044, China
[3]The U.S. Environmental Protection Agency, Research Triangle Park, NC, 27711, U.S.A
[4]Jiangsu Provincial Collaborative Innovation Center for Climate Change, School of Atmospheric Science, Nanjing University, Nanjing, 210023, China
[5]Nicholas School of the Environment, Duke University, Durham, NC, 27710, U.S.A
[6]Department of Health and Environmental Sciences, Xi'an Jiaotong-Liverpool University, Suzhou, 215123, China

*Correspondence to: Jia Xing (xingjia@mail.tsinghua.edu.cn)*

**Abstract.** Airborne aerosols reduce surface solar radiation through light scattering and absorption (aerosol direct effects, ADE), influence regional meteorology, and further affect atmospheric chemical reactions and aerosol concentrations. Several studies have revealed that the inhibition of turbulence and the increase in atmospheric stability induced by ADE increases surface primary aerosol concentration, but the pathway of ADE impacts on secondary aerosol is still unclear. In this study, the two-way coupled WRF-CMAQ with integrated process analysis was applied to explore how ADE impacts secondary aerosol formation through changes in atmospheric dynamics and photolysis processes. Meteorological and air quality fields in Jing-Jin-Ji area (denoted JJJ, including Beijing, Tianjin and Hebei Province in China) in January and July 2013 were simulated to represent winter and summer conditions, respectively. Two pathways of ADE impacts on aerosol concentration, i.e., photolysis modification and atmospheric dynamics modification were estimated separately through scenario analysis. The results show that solar radiation is the restricting factor in winter, and the formation of sulfate is sensitive to the perturbation of solar radiation. While in summer, availability of gaseous precursors primarily dictates the levels of secondary aerosol concentrations. ADE through the attenuation of photolysis inhibits secondary aerosol formation during winter and promotes secondary aerosol formation during summer. The seasonal differences are attributed to change of effective actinic flux in winter and summer determined by aerosol optical depth, solar zenith angles, and single scattering albedo. ADE through dynamic processes is the dominant process influencing surface secondary aerosol formation due to the accumulation of gaseous precursors. Different from sulfate, the surface layer is a net-source of nitrate during winter but a sink during summer. Therefore, ADE promotes nitrate accumulation in winter and reduces nitrate accumulation in summer.



# 1 Introduction

Aerosols have long been recognized as a major source of uncertainty in the climate system due to their interaction with solar radiation and clouds (Carslaw et al., 2013;Koch and Del Genio, 2010;Ramanathan et al., 2001;Rosenfeld et al., 2014). In addition to the cooling or warming effect on global average temperature, aerosols also modulate regional weather due to spatial-temporal heterogeneity in their distributions and further deteriorate air quality indirectly (Ding et al., 2013;Wang et al., 2013;Huang et al., 2016;Ren et al., 2021;Wang et al., 2018b;Wang et al., 2018a;Huang and Ding, 2021;Wang et al., 2015).

Studies in recent decades have revealed the impact of aerosol direct effects (ADE) on air pollutants (Wang et al., 2014;Xing et al., 2015a;Xing et al., 2016;Ding et al., 2016b;Wang et al., 2013;Wang et al., 2018b;Huang et al., 2018;Wang et al., 2018a;Wang et al., 2015;Hong et al., 2020;Atwater, 1971;Ackerman, 1977;Ramanathan et al., 2001;Wendisch et al., 2008;Grell et al., 2011;Wong et al., 2012;Barbaro et al., 2013). Aerosols reduce solar radiation reaching the ground by scattering and absorption. The aerosol direct radiative forcing is estimated as $-15.7\pm9.0$ W/m$^2$ in 2005 in China (Li et al., 2010).

The reduced solar radiation leads to a decrease in temperature (McCormick and Ludwig, 1967). The surface temperature reduction due to ADE is estimated as -0.72 ℃ (Li et al., 2015). Meanwhile, the absorbing aerosols lead to an increased temperature at higher altitude (Ding et al., 2016b). The opposite trend of temperature change between near-surface layer and higher layer in the planetary boundary layer (PBL) is also supported by the air temperature from observation in Beijing and global meteorological reanalysis (Huang et al., 2018;Huang and Ding, 2021).

Compared to the impact pathways of ADE on primary aerosol through inhibition of PBL development, ADE effects on secondary aerosol are much more complicated, making it difficult to analyze. ADE can affect secondary aerosol by changing vertical/horizontal transport and altering its precursors and reaction rate (Li et al., 2017;Liao et al., 2015;Ding et al., 2016a;Yang et al., 2017). Studies have been conducted to explain the impact of aerosol on atmospheric oxidations through

attenuation. He and Carmichael (1999) illustrate the distinct roles of different types of aerosols on photochemical reaction rate and ozone (O$_3$) concentration. Atmospheric aerosols cause significant attenuation of ultraviolet radiation and affect photolysis rates and species chemical cycles (Deng et al., 2012; Mok et al., 2016). Zheng et al. (2015) show that oxidant concentrations fall dramatically during high aerosol loading in winter, suggesting a reduction in secondary aerosols through gaseous reactions. However, impacts of ADE on secondary particle formation through atmospheric dynamic processes has not been well studied.

Reduced ventilation by ADE will concentrate gaseous precursors thereby changing secondary particle formation in surface and upper layers and indirectly influencing the aerosol concentration. Additionally, since secondary aerosol could either form in upper layers and get transported to near-ground level or form in near-ground level and get transported aloft, the modulation of PBL development due to ADE may either increase or decrease surface-level secondary aerosol concentrations. Detailed physical processes of these impacts on near-surface and tropospheric aerosol burden and their quantification are still needed

as is the relative importance of each pathway and their likely seasonal variation. To gain further insight into these pathways, process analysis is conducted in this study.

With the rapid development of economy and the acceleration of urbanization, the air quality in China has been deteriorating in recent decades. In 2010, the population-weighted PM$_{2.5}$ concentration in China was as high as 59 μg/m$^3$. More than 80% of

the residents live in regions where 5-year averaged PM$_{2.5}$ is above the national Class II regional air quality standards (i.e., more than 35 μg/m$^3$) (Apte et al., 2015). In 2013, annual-averaged PM$_{2.5}$ concentrations across 74 key cities in China ranged from 26 to 160 μg/m$^3$, with many locations far exceeding China's air quality standard. The number of premature deaths due to exposure to PM$_{2.5}$ in China is estimated to be more than 1 million for 2010 conditions (Wang et al., 2017;Lim et al., 2012;Apte et al., 2015). In recent decades, extreme air pollution events have occurred frequently across China (Wang et al., 2018a).

Understanding the causes of heavy pollution incidents is needed for developing effective pollution control measures in China. To provide an insight into these questions, this study analyzes the contribution of each pathway for secondary inorganic aerosols. The diurnal and seasonal variations in these pathways are also explored. Investigation on the influence of ADE on atmospheric pollution will provide important guidance for understanding the cause of atmospheric pollution and developing effective control strategies.



## 2 Method

The overall modeling methodology for the study is detailed previously in (Xing et al., 2017) and is briefly summarized here. In this study, the two-way coupled WRF-CMAQ meteorology-chemistry-transport model (Wong et al., 2012) was used to simulate the ADE impacts. Meteorology was simulated by the Weather Research and Forecasting Model (WRF) version 3.4 developed by the National Center for Atmospheric Research (NCAR). Meteorological input data were the National Environmental Prediction Center (NCEP) / NCAR reanalysis data. Pleim-Xiu module (Pleim and Xiu, 2003;Pleim and Gilliam, 2009) was used as land surface scheme. NCEP Automated Data Processing (ADP) global surface and upper-air observation data were carried out for four-dimensional Data assimilation (Grid FDDA). The parameters of other physical processes in the model are MODIS land use type, RRTMG radiation parameterization scheme and ACM2 PBL model. The air quality model used in this study was the Community Multiscale Air Quality Modeling System (CMAQ) of version 5.0.1, developed by the Environmental Protection Agency of the United States. In our previous papers, we have detailed and fully evaluated the model (Xing et al., 2015a;Xing et al., 2015b;Wang et al., 2014;Xing et al., 2017). The comparison of simulated and observed $PM_{2.5}$ concentration is shown in Fig. S1 in supplemental information. Gaseous species and aerosols were simulated by using Carbon Bond 05 (CB05) gas-phase chemistry (Sarwar et al., 2008) with AERO6 aerosol module (Appel et al., 2013). The BHCOAT coated-sphere module (Bohren and Huffman, 1983) was used to simulate aerosol optical properties based on simulated aerosol composition and size distribution (Gan et al., 2015). The gridded emission inventory, initial and boundary conditions used in this study were consistent with our previous studies (Wang et al., 2011;Zhao et al., 2013b;Zhao et al., 2013a;Wang et al., 2014).

Figure 1 shows the modelling domain, which covers most of China and surrounding portions of East Asia, discretized with a 36 km × 36 km grid resolution. CMAQ and WRF both use 23 vertical layers. January 1st to 31st and July 1st to 31st in 2013 was selected to represent winter and summer conditions, respectively. Each simulation was also preceded by a 7-day spin-up period. Jing-Jin-Ji area (denoted JJJ), including Beijing, Tianjin and Hebei Province in China, were selected for the analysis. In this study, observation data across China from the China National Urban Air Quality Real-time Publishing Platform was used to evaluate the model performance. The validation results were shown as Fig. S1 to Fig. S4 in supplemental information.

Following our previous analyses (Xing et al., 2017), three scenarios were simulated, including 1) the baseline simulation (denoted SimBL) in which no aerosol did not change photolysis rates or dynamics were considered, 2) the simulation (denoted SimNF) in which aerosol only affects photolysis rates, and 3) the simulation (denoted SimSF) in which aerosol feedbacks were considered through both photolysis and dynamic processes. The differences between the simulations of SimNF and SimBL was used to present the ADE impacts through photochemistry process (ADEP, denoted Photolysis in the figures). Similarly, the differences between the simulations of SimSF and SimNF was used to estimate the ADE impacts through dynamic process (ADED, denoted Dynamics in the figures). The combined ADE impacts due to both photolysis and dynamics (denoted ΔTotal) were estimated from the differences between the simulations of SimSF and SimBL.

To further explore the impact, Process Analysis (PA) technology (Gipson and Young, 1999) was applied in the simulation of WRF-CMAQ (Xing et al., 2011). The Integrated Process Rates (IPRs) quantify the hourly tendencies from six major modelled atmospheric processes shaping the simulated aerosol concentrations. These process tendencies represent the dominant sinks or sources and include aerosol process (denoted AERO), cloud processes (i.e., the net effect of cloud attenuation of photolytic rates, aqueous-phase chemistry, et al., denoted CLDS), dry deposition (denoted DDEP), horizontal advection (denoted HADV), horizontal diffusion (denoted HDIF), vertical advection (denoted ZADV), and vertical diffusion (denoted VDIF). We combined VIDF, ZADV, and DDEP as vertical transport (VTRN) and combined HDIF and HADV as horizontal transport (HTRN).

## 3 Results and Discussions

The perturbation of ADE on solar radiation and PBL is presented as Fig. 2 and Fig. 3, respectively. As shown in Fig. 2, ADE reduces solar radiation reaching the ground. The daily maximum reduction occurs at noon, with a mean value of 70 W/m$^2$ and 40 W/m$^2$ in January and July, respectively. Decreased solar radiation weakens surface turbulence and reduces the daily





maximum PBL height. Figure 3 illustrates that the impact of ADE on monthly mean PBL height shows a unimodal distribution in January and bimodal distribution in July. PBL height is reduced mostly in the afternoon. The daily average reduction in January and July is about 70 m and 30 m, respectively. Meanwhile, the daily maximum PBL heights are about 500 m and 1500 m in January and July, respectively. It indicates that the change of PBL height is more significant in January.


Further, we investigated the impact of ADE on the vertical profiles of secondary aerosol concentrations through dynamic pathway and photolysis, as shown in Fig. 4. To illustrate the pathways of ADE impacts on aerosols, we choose sulfate and nitrate as the main study subjects and elemental carbon (EC) as a reference. ADE weakens turbulence induced vertical mixing in both January and July, which leads to primary pollution being trapped in lower layers. In January, EC concentrations increase
by 7.5% due to ADE in near surface layer and decreases by 5% at 600 m. For the case of secondary pollution, ADE affects sulfate predominantly through modulation of photolysis rates, which leads to a decrease of sulfate formation rate in all layers. The reduction rate is about 3% on average in the near surface layer. Dynamic processes lead to an increase in sulfate concentration in the near surface layer and a decrease of sulfate concentration above 300 m. These two processes combined contribute to 7.5% reduction of sulfate at 900 m, which is the strongest affected layer in terms of sulfate concentration. It may
be noted that the reason for the noted strongest impact aloft for sulfate is that $SO_2$ (precursor for sulfate) is emitted from tall stacks. In July, the ADED is the key process altering sulfate concentration. The strongest impact is at 1100 m. Compared to its feedback effects on atmospheric dynamics, ADE barely changes sulfate concentration through photolysis in July. This is because, in spite of attenuation due to the presence of aerosols, solar radiation is abundant in July. Meanwhile, the change of solar radiation due to ADE is not as strong as in January. Traditionally, the pathway through changing of actinic flux is
emphasized, but the pathway through dynamic process and further change of gaseous precursors is barely mentioned. Our results indicate that ADE affecting sulfate formation through dynamic pathway is a metric of equal, or greater, importance than that of photolysis pathway in both summer and winter.

The analysis on how ADE affects secondary aerosol concentration is complicated. To provide insight into how ADE affects
surface sulfate concentration, the vertical distribution of the IPRs response to ADE is presented as Fig. 4. It shows that during winter, the contribution of ADED on the sulfate is at a similar level compared to that of ADEP. The influence of ADEP in January is mainly reflected in the reduction of sulfate formation (AERO, Fig. 4a red). This effect occurs at almost all altitudes and greater at lower altitudes. ADED is mainly reflected in the weakening vertical transport (VTRN) of sulfate concentration (Fig. 4b purple) caused by shallower PBL. Further, the weakening VTRN caused by ADED results in an increase of sulfate
concentration below 500m and decreased sulfate concentration above 500m decreases. The dividing point is at a similar altitude to daily max PBL height. Moreover, the dynamic path barely changes the AERO process (Fig. 4b red). It implies that the ADED affects sulfate concentration mainly by trapping sulfate in the near surface layer rather than changing $SO_2$ concentration and sulfate formation. The superposition of photolysis and dynamic pathway (Fig. 4c) leads to an increase in the overall concentration of sulfate in the boundary layer and a decrease in the concentration in free atmosphere. Overall, both ADED and
ADEP make apparent contribution to the sulfate concentration (Fig. 4b) in January (Fig. 4a). In contrast, the contribution of ADED in July (Fig. 4e) is much higher than that of ADEP (Fig. 4d) to the change in sulfate concentration. The photolysis pathway in January is mainly reflected in the increase of AERO. The dynamic path is mainly reflected in the weakening of vertical transmission in the boundary layer, the enhancement of aerosol generation as well as liquid phase reactions.

In the seasonal comparison of the influence of ADE on the AERO process, there are three interesting points. First, the influence of changes in the photolysis pathway on aerosol formation is negative in winter and positive in summer. This is mainly due to the different effects of light absorption and scattering on aerosols and surface albedo. Usually, scattering aerosol increases the effective optical path length and raise the total actinic flux in the atmosphere as a whole, while absorbing aerosol decreases the actinic flux in the layer below, compared with aerosol-free scenario (Dickerson et al., 1997;Herman et al., 1999). The
influence of aerosol on the photochemical reactions also varies with single scattering albedo (SSA). A low SSA value (strong absorption) tends to inhibit the photochemical reaction, while a high SSA tends to promote the photochemical reaction. Moreover, such impact varies with altitudes and aerosol loadings. Forward scattering increases actinic flux of the layer below, given that the diffuse light increases the effective optical path length. Backward scattering increases the actinic flux of the layer above aerosol but decreases the actinic flux under the aerosol layer. Thus, the ground-level actinic flux will depend on
aerosol loading and vertical distribution. The factors impacting actinic flux include but are not limited to single scattering



albedo, aerosol loading (aerosol optical depth, $\tau$) and solar zenith angle ($\theta$). Higher effective optical depths ($\tau /\cos \theta$, a variable to represent aerosol loading) attenuate direct solar radiation and increase the diffusion of solar radiation. Thus, this impact will be more significant at high $\theta$ (Dickerson et al., 1997;He and Carmichael, 1999) and high $\tau$. In January, the average AOD reaches to 2.5, much higher than the annual average level (Bi et al., 2014). Coal combustion and biomass burning, especially for residential heating, leads to high levels of black carbon, which results in low SSA. High aerosol loading, low SSA, and low solar zenith angle together lead to decreased actinic flux in near-ground layers, due to ADE. On the contrary, low aerosol loading, high SSA and high solar zenith angle together lead to increased actinic flux in near-ground layers in July. Second, the impact of ADE through dynamic pathway is at similar level to that of photolysis pathway in January and acts as dominant factor of ADE in July. ADE through dynamic pathway changes sulfate by trapping sulfate at near surface level in January and promotes sulfate formation in July. Moreover, the dynamic pathway has a positive effect for aerosol formation in summer, and its contribution is much higher than the photochemical path, especially in the boundary layer. These three points all indicate that solar radiation is the restricting factor in winter, and the formation of sulfate is sensitive to the perturbation of solar radiation. In summer, solar radiation is abundant and sulfate formation is primarily limited by availability of gaseous precursors. The ADE mainly alters precursor concentration through dynamic process, eventually affect sulfate formation.

To further investigate the impacts of ADE on atmospheric chemistry, we examined the changes in concentrations of oxidants, defined as the sum of O, $O_3$, $NO_2$, $NO_3$, $N_2O_5$, $HNO_3$, peroxynitric acid, alkyl nitrates and peroxyacyl nitrates. The modification of atmospheric oxidants by ADE also shows solar radiation control in January and gaseous precursor control in July, which further support the above discussions. The atmospheric oxidation is an important factor related to secondary aerosol formation. Figure 5 provides an overview of the atmospheric oxidation change induced by ADE. In January, ADEP is the dominant process to impact atmospheric oxidation. It leads to a decrease of oxidants in the layer below 1 km and an increase in oxidants above it. ADED slightly raises oxidation near ground and exhibits little impact on layers above 500 m. In July, both dynamic and photolysis pathways are important. ADEP increases atmospheric oxidants in all layers. The height of strongest effect is about 600 m. ADED amplify near-surface atmospheric oxidants but reduces atmospheric oxidants above 600 m.

The impact of ADEP shows a clear relation with effective optical path length, shown as Fig. 6. As discussed in section 3.2, except SSA, promotion or inhibition of secondary aerosol formation by ADEP is impacted by AOD and effective optical path length (mainly determined by solar zenith angle). Fig. 6a shows ADEP inhibits surface sulfate formation in the daytime, since aerosol with high SSA and long optical path length due to large solar zenith value in January reduce the actinic flux. In July, sulfate formation is inhibited in early morning and late afternoon. ADEP slightly promotes sulfate formation at noon. This result could be explained by scattering aerosol. As described in section 3.2, scattering aerosol reduces direct solar radiation and increases diffuse solar radiation. Diffusion solar radiation is more easily utilized in photolysis due to the large effective optical path length. High SSA, which indicates high fraction of scattering aerosol in summer, together with strong solar radiation at noon leads to the promotion of photolysis. Along with the strong ADED effects, sulfate formation is promoted from 10:00 to 15:00 in summer.

The ADE impacts on nitrate are also limited by solar radiation in winter and gaseous precursors in summer. Figure 8 displays that ADEP reduces nitrate formation in the near ground layer in January. Similar amount of nitrate is affected by ADEP and ADED in the near ground layer. ADED is dominant in the upper layers in January and in all layers in July. Like sulfate, surface nitrate formation (Fig. 8a red) is reduced by ADEP. ADEP barely changes the nitrate formation in July. The impact of ADE on transport is more complicated for nitrate and differs according to the transport direction. The main vertical transport direction is opposite in January and July. As shown in Fig. 7, nitrate is mainly formed at high altitude due to the lower temperature in January and is entrained to the surface as the PBL grows, which is also proved by previous study (Huang et al., 2021;Curci et al., 2015). Thus, it shows a positive VTRN and negative AERO in the near ground layers. AERO is the main sink in the near ground levels. Nitrate related process in the layers above 500 m presents opposite results compared to ground levels. AERO is the main source and VTRN is the main sink. On the contrary, the transport direction is bottom-up in July. Same as sulfate, ADE through dynamics affects nitrate concentration through two major pathways, i.e., vertical transport (shown in Fig. 8b as purple) and precursor concentration (shown in Fig. 8b as red). Except surface layer in January, ADE always inhibits the transport of nitrate in January. The transport direction in January is top-down. VTRN in near ground layers is negative (shown in shown in Fig. 8b as purple), which indicates that the transport of nitrate from upper layer to near ground





layer is inhibited. Similar to sulfate, the accumulation of gaseous precursors induces an increase in nitrate formation. It further increases the absolute amount of nitrate transport.

## 4 Conclusions

In addition to directly deteriorating air quality, aerosol diminishes solar radiation due to light scattering and absorption thereby influencing regional meteorology and thus further modulating air quality. The impact of ADE on secondary aerosol is more complicated than primary aerosol. This study quantified the impacts of ADE on $PM_{2.5}$ using the two-way online coupled meteorology and atmospheric chemistry model (WRF-CMAQ) with integrated process analysis. The main pathways through which ADE affect aerosol concentrations were examined. The key conclusions are summarized below: 1) ADE reduces solar radiation and decreases PBL height, trapping aerosol in near ground layers. Including ADE improves the model performance for simulating $PM_{2.5}$ and its components. The mechanism of ADE impacts on secondary aerosol is more complicated than primary aerosol. 2) solar radiation is the restricting factor in winter, and the formation of sulfate is sensitive to the perturbation of solar radiation. During summer, solar radiation is abundant for both photolysis and dynamic processes and the gaseous precursor availability limits sulfate formation and abundance. 3) ADEP inhibits secondary aerosol formation during winter in the JJJ region and promotes secondary aerosol formation in July. The differences are attributed to the alteration of effective actinic flux affected by AOD, solar zenith angle and SSA. 4) ADED traps sulfate and nitrate in the surface layer which increases secondary aerosol concentration in winter. Meanwhile, the impact of ADED is mainly reflected in the increase of gaseous precursors concentrations, subsequently enhancing secondary aerosol formation. Further, the influence of ADED is associated with transport directions. Turbulence is weakened by the reduction of solar radiation and enhanced temperature inversions. The major transport direction of sulfate and its gaseous precursors is bottom-up in both winter and summer. Thus, ADE induced weakened atmospheric ventilation traps sulfate and its gaseous precursors in the near ground layers. Different from sulfate, surface layer acts as the source for nitrate in winter but sink in summer. Therefore, ADED promotes nitrate accumulation in winter and reduces nitrate accumulation in summer.

**Author contributions:** J.D.W, J.X and J.P.W. wrote the manuscript with inputs from all co-authors. J. X. and J.D.W. performed the simulation and analyzed the data. D.D. and D. W. supported the model configuration and simulation. S.W., R.M., Y.Z., J.E.P., X.C., J.J., P.Z., S.S., Y.J., and J.H. discussed the results and commented on the manuscript.
**Data availability:** Model outputs are available upon request from the corresponding author.

### Acknowledgements

This work was supported in part by National Science Foundation of China (42075098 & 41907190), National Key R&D program of China (2018YFC0213805). This work was completed on the "Explorer 100" cluster system of Tsinghua National Laboratory for Information Science and Technology.
**Competing interests:** The authors declare that they have no conflict of interest.
**Disclaimer:** The views expressed in this paper are those of the authors and do not necessarily represent the view or policies of the U.S. Environmental Protection Agency.

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





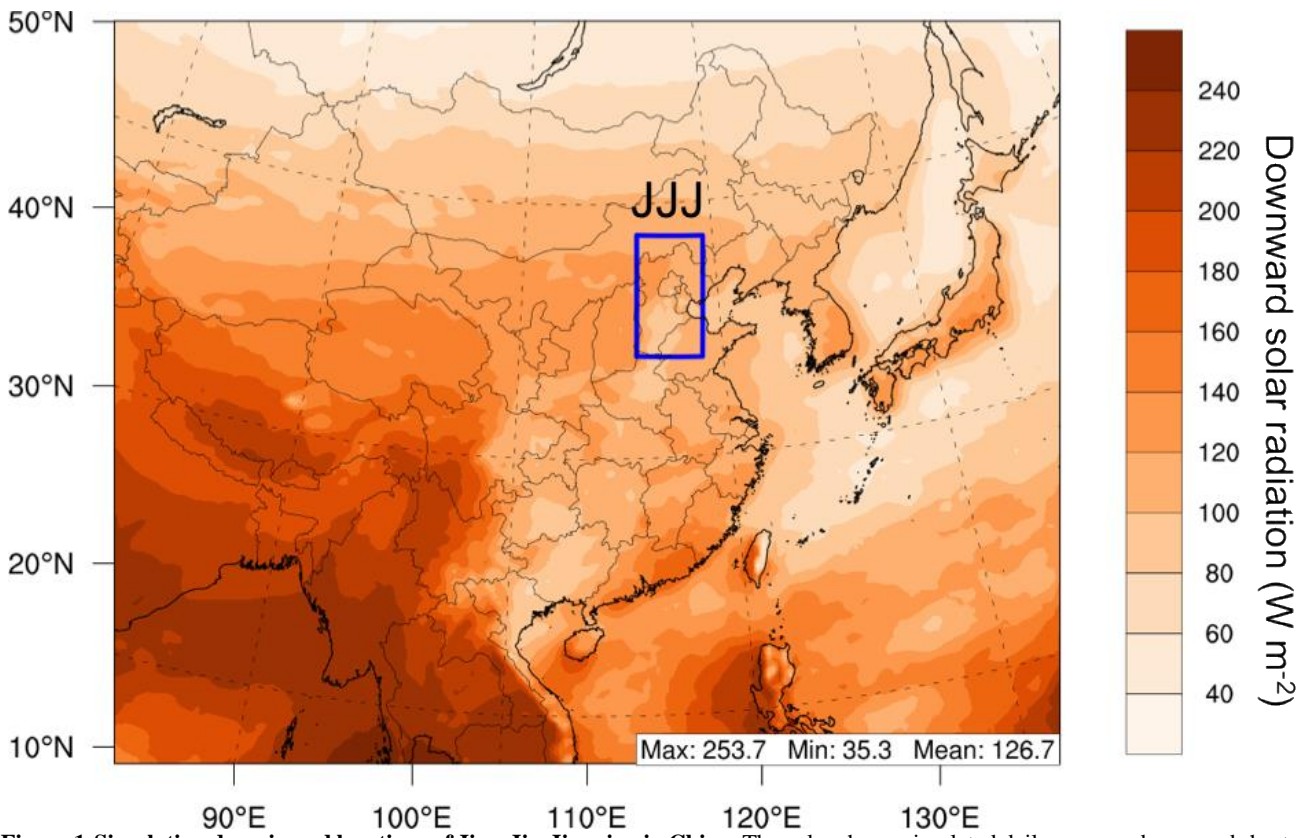

**Figure 1:Simulation domain and locations of Jing-Jin-Ji region in China.** The color shows simulated daily average downward shortwave solar radiation (SWDOWN) at bottom in January, 2013





**Figure 2: Diurnal variances of SWDOWN (a and c) and the impact of ADE on SWDOWN (b and d), in January and July 2013**. The
central rectangle spans the first quartile to the third quartile. The segment and red dot inside the rectangle show the median and mean value,
respectively. The whiskers above and below the box extend to the highest and lowest values.






**Figure 3: Diurnal variances of Planetary Boundary Layer (PBL) height (a and c) and the impact of ADE on PBL height (b and d), in January and July, 2013.** The central rectangle spans the first quartile to the third quartile. The segment and red dot inside the rectangle show the median and mean value, respectively. The whiskers above and below the box extend to the highest and lowest values.

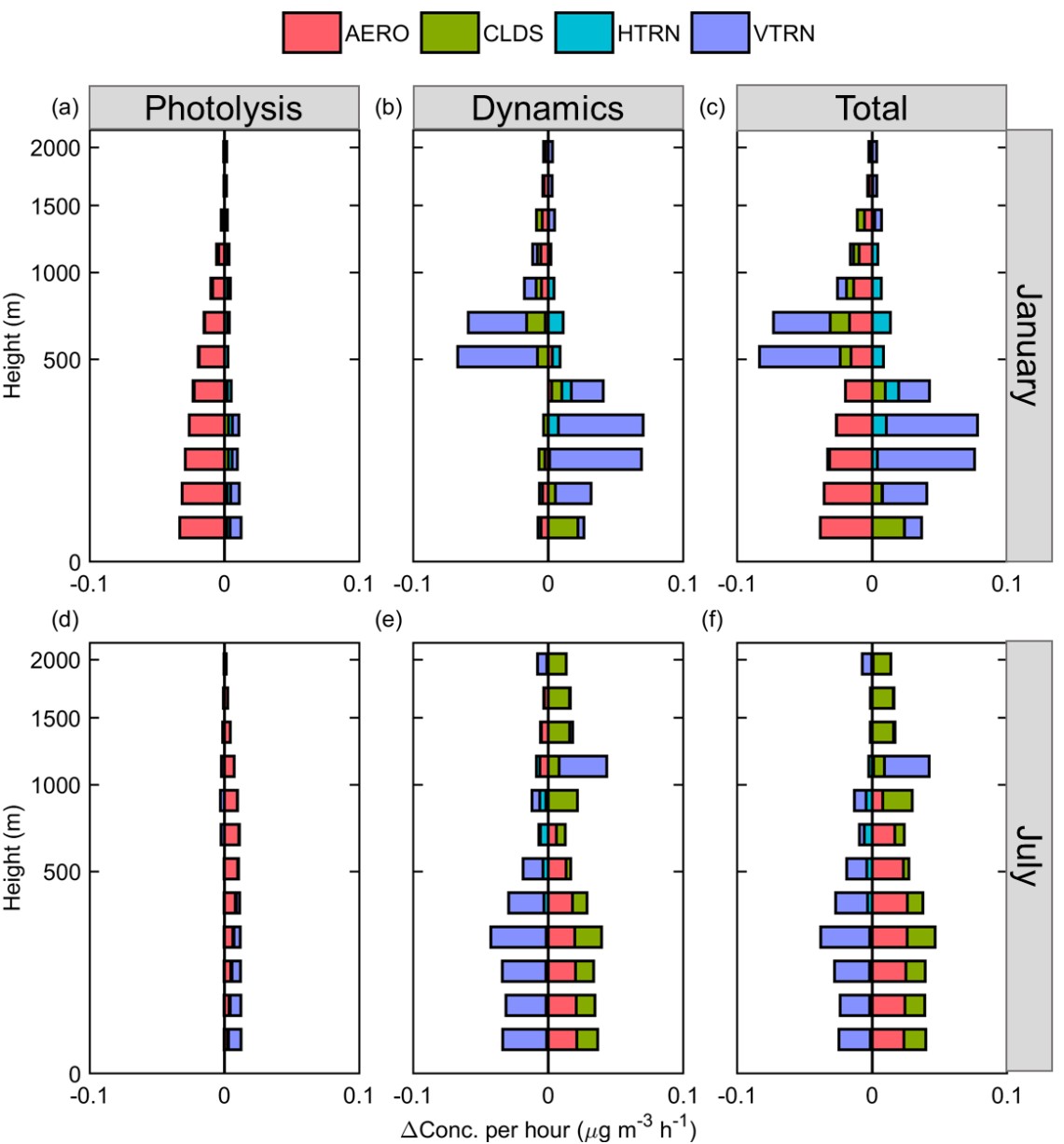

**Figure 4: Vertical distribution the responses of main process of sulfate to ADE in Jing-Jin-Ji(JJJ) region in January (a b c) and July (d e f).**





**Figure 5: Vertical distribution of ADE impact on mean total oxidation concentration.** The red line and shadow show the medium value and 25th to 75th percentiles, respectively.



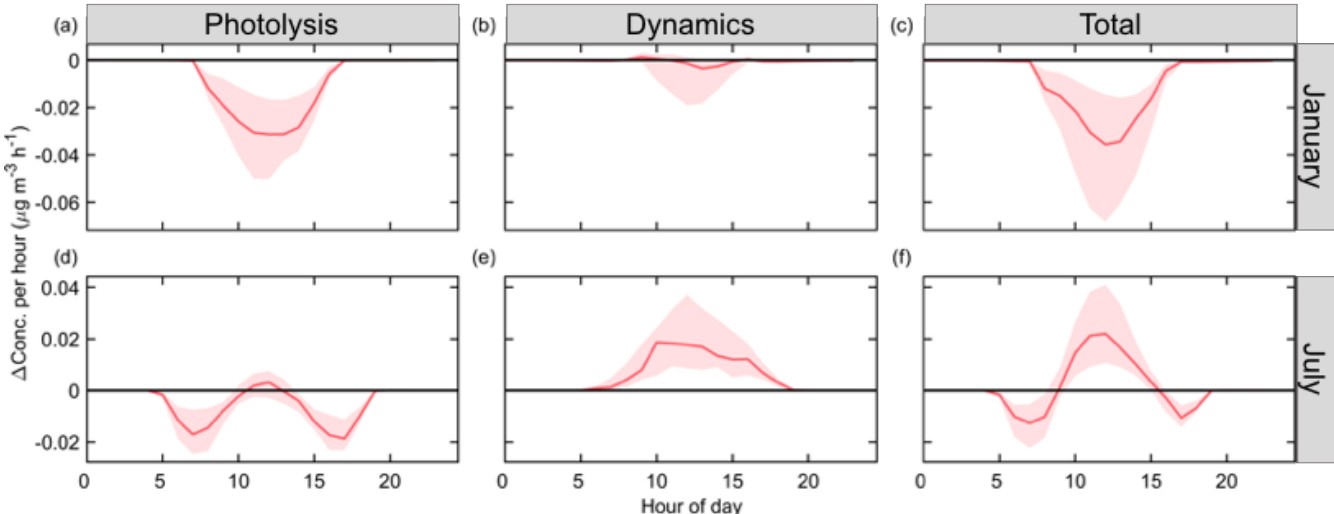

**Figure 6**: **Diurnal variances of ADE impact on AERO in January and July.** The red line and shadow depict the medium value and 25th to 75th percentiles, respectively.



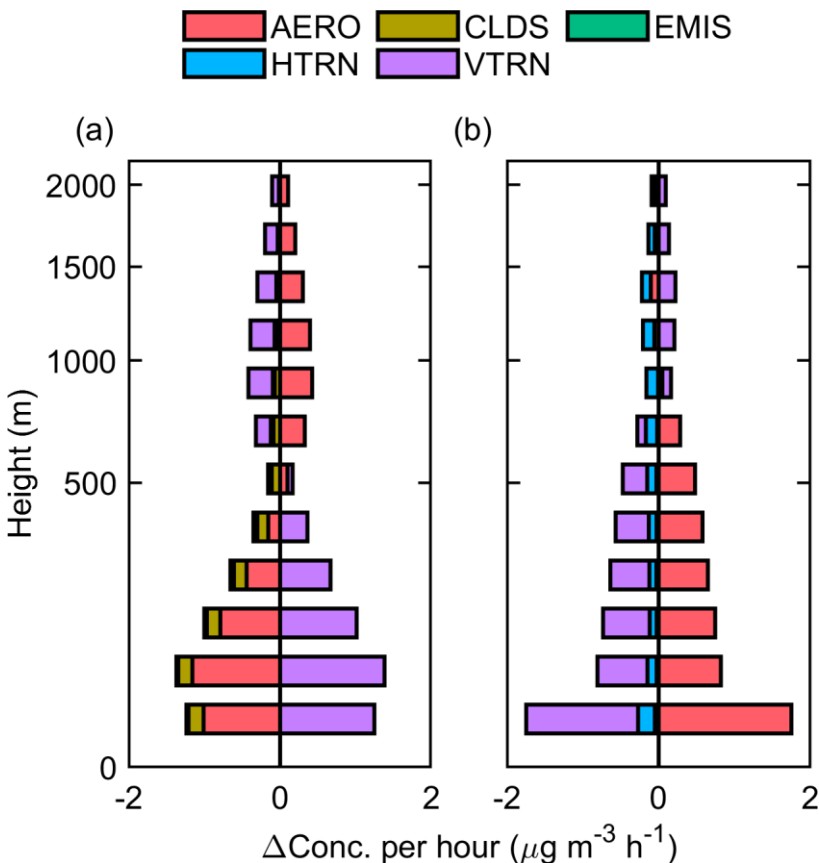

**Figure 7: The monthly mean of vertical distribution of main process of nitrate in January (a) and July (b).**





**Figure 8: Vertical distribution the responses of main process of nitrate to ADE in Jing-Jin-Ji(JJJ) region in January (a b c) and July (d e f).**