# Peer review of "The pathway of aerosol direct effects impact on secondary inorganic aerosol formation"

_Atmospheric Chemistry and Physics, 2021_

## Author Comment (AC1)

*This short and concise study investigates the different pathways of aerosol direct effects impact on sedondary inorganic aerosol formation between winter and summer. The results show that solar radiation is the restricting factor in winter, and the formation of sulfate is sensitive to the perturbation of solar radiation. While in summer, availability of gaseous precursors primarily dictates the levels of secondary aerosol concentrations. The findings are valuable and interesting to the science community. Thus, I would like to recommend its acceptance for publication after necessary minor revisions.*

**Response:** We appreciate the reviewer's kind effort and constructive comments. We have implemented all suggestions for improvement in the revised manuscript. Please kindly find our point-by-point responses listed below. The reviewer's comments are in Italic and blue followed by our responses and revisions.

*General comments*
*There are some writing issues with several figured out in the following comments.*
*Line 37-40, aerosol effects on regional weather is not due to their spatio-temporal distribution, but due to the radiative effect or impact on cloud properties. In addition, three studies about the aerosol impacts on regional weather are suggested here, Sun and Zhao (2021, doi: 10.5194/acp-21-16555-2021) for aerosol impacts on precipitation time, Zhao et al. (2018, 2020, doi: 10.1093/nsr/nwz184, doi: 10.1029/2018GL079427) for aerosol impacts on weather over Tibetan Plateau and western pacific.*

**Response:** Thanks for the suggestion. We have rewritten the sentence and added related discussion of the references.

Line 36, Page 2 in revised manuscript
"*It perturbs the Earth's energy budget through aerosol direct effects (ADE) by direct scattering and absorbing shortwave and longwave radiation and indirect effects via interaction with cloud.*"

Line 38, Page 2 in revised manuscript
"*Besides the climatic effects, studies in recent decades have revealed that it alters regional weather (Sun and Zhao, 2021; Zhao et al., 2018, 2020).*"

*Line 40, What do the authors mean "aerosol direct effect on air pollutants"? The authors might provide a defidition for the aeroosl direct effects (ADE).*

**Response:** Thanks. We have revised this sentence with definition for aerosol direct effects.

Line 36, Page 2 in revised manuscript
"*It perturbs the Earth's energy budget through aerosol direct effects (ADE) by direct scattering and absorbing shortwave and longwave radiation and indirect effects via*

*interaction with cloud.*"

Further, we explain the detail mechanisms of how ADE affects air pollutants.

Line 43, Page 2 in revised manuscript

"*Absorption and scattering of aerosols reduce the solar radiation reaching to the ground which lower the surface temperature (McCormick and Ludwig, 1967;Li et al., 2015; Yang et al., 2016b, 2018). Meanwhile, aerosols can heat up the air in upper-layer with the presence of absorbing components (black carbon, brown carbon and dust)(Ding et al., 2016b; Huang et al., 2018;Wang et al., 2018a). Such controversial effects modify the vertical temperature profile and suppress the development of PBL, resulting in accumulation of pollutants in near-surface layer and aggravation of atmospheric pollution (Huang and Ding, 2021).*"

*Line 43-44, this is true, which could further reduce the planetary boundary layer height and near surface wind speed, resulting in further heavier aerosol pollution, as indicated by Yang et al. (2016, doi:10.1002/2015JD024645).*

**Response:** We agree. We have added the reference and related discussion.

Line 38, Page 2 in revised manuscript

"*Besides the climatic effects, studies in recent decades have revealed that it alters regional weather (Sun and Zhao, 2021; Zhao et al., 2018, 2020). Airborne aerosols can alter planetary boundary layer (PBL) development (Atwater, 1971;Ackerman, 1977;Ramanathan et al., 2001;Wendisch et al., 2008;Grell et al., 2011;Wong et al., 2012;Barbaro et al., 2013; Wang et al., 2013) and further deteriorate air quality, which is defined as aerosol-PBL interactions. (Ding et al., 2013; Wang et al., 2014; Xing et al., 2015a;Xing et al., 2016; Wang et al., 2018b;Wang et al., 2015; Yang et al., 2016a; Hong et al., 2020).*"

*Line 44-45, regarding the aerosol solar radiative cooling effect, a few references might be helpful, such as Yang et al. (2016,2018, doi:10.1002/2016JD024938, doi:10.1016/j.atmosres.2018.04.029). In addition, it might be not necessary to indicate the values here since they should vary with time and location.*

**Response:** We thank for the suggestions. We have removed the detail values and rewritten the paragraph. The related references are added to support our statements.

Line 43, Page 2 in revised manuscript

"*Absorption and scattering of aerosols reduce the solar radiation reaching to the ground which lower the surface temperature (McCormick and Ludwig, 1967;Li et al.,*

*2015;Yang et al., 2016b, 2018). Meanwhile, aerosols can heat up the air in upper-layer with the presence of absorbing components (black carbon, brown carbon and dust)(Ding et al., 2016b; Huang et al., 2018;Wang et al., 2018a;). Such controversial effects modify the vertical temperature profile and suppress the development of PBL, resulting in accumulation of pollutants in near-surface layer and aggravation of atmospheric pollution (Huang and Ding, 2021).*"

*Line 52, it might be useful to define "secondary aerosol" first. Also, personally, I would more prefer using "seondary formed aerosol".*

**Response:** Thanks for the comment. We have added the explanation of secondary aerosol in revised manuscript.

Line 50, Page 2 in revised manuscript

"*Compared to the impact pathways of ADE on primary aerosol through inhibition of PBL development, ADE effects on secondary aerosol, which is formed in the atmosphere through atmospheric reaction, are much more complicated.*"

*Line 55, "illustrate" should be "illustrated"?*

**Response:** Thanks for your suggestion. We have revised it in manuscript.

*Line 57, "show" should be "showed"?*

**Response:** Thanks for your comment. We have revised it in manuscript.

*Line 59, "has" -> "have"*

**Response:** Thanks. We have revised it in manuscript.

*Line 69-74, this should be the air quality status of the past, not current. However, as indicated by Fan et al. (2020, doi: 10.1016/j.atmosenv.2019.117066) and Zhang et al. (2020, doi: 10.1007/s13143-019-00125-w), the air quality in China has improved significantly owing to the strict control acts in China. This fact should be acknowledged.*

**Response:** Yes. We agree. We have added the following description and references in revised manuscript.

Line 74 , Page 2 in revised manuscript

"*The air quality in China has improved significantly since 2013, owing to the strict control acts in China (Fan et al., 2020; Zhang et al., 2020)*"

*Line 81, it should by "in Xing et al. (2017)"*

**Response:** Thanks for your suggestion. We have revised it in manuscript.

*Line 87-88, I belive the "other physical processes" should be more than what described here. The authors might slightly modify the description.*

**Response:** Thanks. We have added more description of physical schemes in revised manuscript.

Line 86, Page 3 in revised manuscript

"*The Pleim-Xiu land surface model (Pleim and Xiu, 2003; Pleim and Gilliam, 2009), associated with Asymmetric Convective Model of version 2 (ACM2) PBL scheme was used in this study. MODIS land-use type was chosen. RRTMG radiation parameterization scheme was used for shortwave and longwave radiation treatment. The Morrison 2-Moment microphysics scheme and Kain-Fritsch cumulus scheme were used in this study.*"

*Line 99, what are the vertical resolutions for the two model simulations within boundary layer heigtht?*

**Response:** Thanks for the comment. It is kind of hard to count for the detail layers within boundary layer (PBL) height, due to its variations over time. But 8 layers are set under 1 km in both WRF and CMAQ in our study. We modified the following description in revised manuscript.

Line 102, Page 3 in revised manuscript

"*WRF and CMAQ both use 23 vertical layers, in which 8 layers are set under 1 km to better describe the boundary layer processes.*"

*Line 101-103, How reliable are the observation data from this platform? Are there any scientific studies based on the dataset that can serve as supporting references?*

**Response:** Thanks for the comment. The observation data was obtained from the China National Urban Air Quality Real-time Publishing Platform supported by Ministry of Ecology and Environment, China. Their calibrations and quality controls are guaranteed by the China National Environmental Monitoring Center (CNEMC). Several studies have used this database to validate the air quality model simulation (e.g. Zhang et al., 2021; He et al., 2020). We have modified the description in revised manuscript.

Line 105, Page 3 in revised manuscript

"*In this study, observation data from the China National Urban Air Quality Real-time Publishing Platform supported by Ministry of Ecology and Environment, China was used to evaluate the model performance.*"

*Line 110, "was" -> "were"*

**Response:** Thanks for your suggestion. We have revised it in manuscript.

*Line 114-121, a breif description about the PA technology and IPRs is necessary for readers to understand.*

**Response:** We agree. A brief description is added in revised manuscript.

Line 120, Page 3 in revised manuscript

"*Eulerian chemistry transport model simulates air pollution concentration by solving transport partial differential equations. A series of physical and chemical processes is calculated to determine the changes in species concentration at each timestep. Based on the properties of linear equation, PA could estimate the accumulated effects of each process. The Integrated Process Rates (IPRs) quantify the hourly tendencies from six major modelled atmospheric processes shaping the simulated aerosol concentrations.*"

*Line 124-125, I wonder if these are monthly average values including days with clouds? Or simply, are these values monthly averages for clear skies? How did the authors exclude the clouds?*

**Response:** Thanks for your comments. The monthly average values in this study include days with clouds. The aerosol direct effect under all skies is considered in this study. We didn't exclude the clouds because we'd like to investigate the ADE influences on aerosol under the condition that close to reality with clouds being involved. The PBL height is affected by both cloud and aerosol under cloud condition. Further, the cloud effects exist in scenarios with/without considering aerosol direct effect. Thus, the difference between scenarios shows aerosol effects under cloud condition.

*Line 127-129, Similarly, are the results for all skies or clear skies only?*

**Response:** Thanks for your comment. The aerosol direct effect under all skies is considered in this study.

*Line 168, "raises"?*

**Response:** Thanks. We have revised it in manuscript.

*Line 176, the authors can simply describe "effective optical depths" as "optical paths" without further definition or explanation.*

**Response:** Thanks for your suggestion. We have revised it in manuscript. Further, following the suggestion of reviewer 2 and for better understanding, this part is moved to SI.

*Line 177-178, It is not robust to say "this impact will be more significant at high tau" since it actually depends: when tau is not too high, the diffuse increase with tau; however, when tau is large enough, the diffuse radiation will decrease with tau.*

**Response:** Thanks for your comment. For the sake of rigor, we have revised the expression of this sentence. Further, following the suggestion of reviewer 2 and for better understanding, this part is moved to SI.

Line 60, Page 3 in SI

"*Higher optical depths attenuate direct solar radiation. Thus, this impact will be more significant at high θ (Dickerson et al., 1997;He and Carmichael, 1999) and high τ.*"

*Line 179-180, This increase to 2.5 is not a common phenomenon that can be observed frequently, thus I would suggest changing "reaches to 2.5" to "reached 2.5".*

**Response:** Thanks. We have revised it in manuscript according to your suggestion.

*Line 189, "affect" -> "affects"*

**Response:** Thanks for your suggestion. We have revised it in manuscript.

*Line 198-199, I would suggest changing "The height of strongest effect is" to "The height with the strongest effect is"*

**Response:** Thanks. We have revised it in manuscript accordingly.

*Line 199, "amplify" -> "amplifies"*

**Response:** Thanks. We have revised it according to your suggestion.

*Line 204, "reduce" -> 'reduces"*

**Response:** Thanks for your suggestion. We have revised it in manuscript.

*Line 210, local time or UTC time?*

**Response:** Thanks. It is local time. We have revised the related part in manuscript.

*Line 218, "studies"?*

**Response:** Thanks for your suggestion. We have revised it in manuscript.

*Line 225, delete one "shown in" from the twos.*

**Response:** Thanks. We have revised it in manuscript.

**References**

He, G., Pan, Y., and Tanaka, T.: The short-term impacts of COVID-19 lockdown on urban air pollution in China, Nat Sustain, 3, 1005–1011, https://doi.org/10.1038/s41893-020-0581-y, 2020.

Zhang, Y., Liu, X., Fang, Y., Liu, D., Tang, A., and Collett, J. L.: Atmospheric Ammonia in Beijing during the COVID-19 Outbreak: Concentrations, Sources, and Implications, Environ. Sci. Technol. Lett., 8, 32–38, https://doi.org/10.1021/acs.estlett.0c00756, 2021.

---

## Author Comment (AC2)

*This manuscript presents a model-based analysis on aerosol-radiation-boundary layer interactions and feedbacks, with a focus on secondary sulfate and nitrate formation under polluted conditions. The topic is original and the paper appears scientifically sound. I have a few, mostly minor, issues to be considered before acceptance of this paper for publication.*

**Response:** We appreciate the reviewer's kind effort and constructive comments. We have implemented all suggestions for improvement in the revised manuscript. Please kindly find our point-by-point responses listed below. The reviewer's comments are in Italic and blue followed by our responses and revisions.

*Scientific issues:*

*main comment concerns the structure of section 3. Now there is three longish paragraph discussing sulfate formation, then two short paragraphs on oxidants and AOD, and finally something about nitrate formation. I wonder whether this is the best way of presenting the results for a reader to easily follow the discussion. Furthermore, there appears to be some unnecessary repetition of text in this section. For example, the relative roles of ADEP and ADED in forming sulfate in summer and winter is discussed in three places (lines 146-147, 159-161, 183-184).*

**Response:** Thanks for your comments. As you suggested, we have modified the section 3 to make a clearer presentation of results for readers to follow. We have added two figures showing ADE impacts on sulfate and nitrate concentration in revised manuscript (Fig. 4 and 7 in revised manuscript), making it easier to discuss the overall ADE impacts than using the figure of IPRs (Fig. 4 and 7 in original manuscript where the sum of every bar represents the overall ADE impact). Also, we have moved the figure of oxidants and related discussion to SI and added more description regarding ADE impacts on nitrate. Furthermore, we have polished the description of sulfate formation part to make it easier to follow.

As for the unnecessary repetition, we have checked the structure of section 3 and found that there are some similar descriptive sentences since the overall pattern and detailed processes are discussed separately where same phenomenon may be mentioned in both paragraphs. We have carefully looked through the section 3 and reorganized the description of results avoiding unnecessary repetition according to your suggestion.

The new Figure 4 and 7 with related revision of text are shown below and in revised manuscript.

Line 140, Page 4
"*As shown in Fig. 4, ADE affects sulfate through both photolysis and dynamics in January, leading to a decrease of sulfate formation rate in all layers...*"

Line 174, Page 4

"*The ADE impacts on nitrate are then investigated. Vertical profile of nitrate affected by ADE is presented in Fig. 7. Overall, ADED makes stronger influence on nitrate concentration than ADEP in both winter and summer. ADEP slightly reduces nitrate concentration near surface in both seasons (Fig. 7a and 7d). As for ADED, it generally lower the nitrate concentration in winter (Fig. 7b) and the largest reduction occurs above PBL (at around 900 m). During summer, ADED exhibits a promotion effect on nitrate especially in near surface layers (Fig. 7e).*"

[Figure]

**Figure 4: Vertical profile of sulfate concentration change to ADE in Jing-Jin-Ji(JJJ) region at noontime in January (a b c) and July (d e f).**

[Figure]

**Figure 7: Vertical profile of nitrate concentration change to ADE in Jing-Jin-Ji(JJJ) region at noontime in January (a b c) and July (d e f).**

*Related to the previous comment, the authors refer to section 3.2 on lines 201 and 206, a section which does not exist. I wonder whether some earlier versions of this paper have had structure different from the current one.*

**Response:** Thanks for your comment. Indeed, section 3 was divided into 3 parts in early version. We have revised the related part in manuscript.

*The list of compounds given on line 192 certainly participate in atmospheric oxidation reactions, but not all of them (e.g. NO2 and HNO3) can be considered as oxidants. Please reword and modify this part of the text accordingly.*

**Response:** Thanks for your suggestion. We use ADE on production rate of reacted OH instead of Total Ox in revised manuscript. Further, we have moved the discussion of oxidants to SI in manuscript as mentioned in above response.

Line 41 Page 2 in SI.
"*To further investigate the impacts of ADE on atmospheric chemistry, we examined the changes in production rates of new reacted OH, shown in Fig S5.*"

*Essentially the same thing is stated on lines 230-231 and 235-236. Please avoid repetition. Also, I would suggest some rewording: ...more complicated than its impact on primary aerosol.*

**Response:** Thanks for your suggestion. As mention in above response, we have checked the results and conclusion part and removed repeated sentences accordingly.

*Minor technical issues:*

*line 48: ... observations...*

**Response:** Thanks for your suggestion. We have revised it in manuscript.

*section 2 title should read "Methods"*

**Response:** Thanks. We have revised it according to your comment.

*line 127: The PBL height ...*

**Response:** We appreciate your comment. We have revised it in manuscript.

*lines 135, 137 and 138: in the near-surface layer*

**Response:** Thanks. We have revised it in manuscript.

*line 185: ... effect on ...*

**Response:** Thanks for your comment. We have revised our manuscript accordingly.

*line 207: diffuse solar radiation*

**Response:** Thanks for your suggestion. We have revised it in manuscript.

*Figure 8 is referred to before Figure 7. Please check out that they are referred to correctly in the paragraph on lines 212-227. If necessary, change the order of figures*

*such that they are referred to in correct numerical order.*

**Response:** Thanks. We have revised it in manuscript according to your suggestion.